# Analyzing the correlation between protein expression and sequence-related features of mRNA and protein in *Escherichia coli* K-12 MG1655 model

**Nhat H.M. Truong[1,2], Nam T. Vo[1,2,3], Binh T. Nguyen[2,4], Son T. Huynh[2,4], Hoang D. Nguyen ©[1,2]***

1 Center for Bioscience and Biotechnology, University of Science, Ho Chi Minh City, Vietnam, 2 Vietnam National University, Ho Chi Minh City, Vietnam, 3 Laboratory of Molecular Biotechnology, University of Science, Ho Chi Minh City, Vietnam, 4 Department of Computer Science, University of Science, Ho Chi Minh City, Vietnam

* ndhoang@hcmus.edu.vn

**Data Availability Statement:** All relevant data are within the paper and its Supporting Information files.

## Abstract

It was necessary to have a tool that could predict the amount of protein and optimize the gene sequences to produce recombinant proteins efficiently. The Transim model published by Tuller *et al.* in 2018 can calculate the translation rate in *E. coli* using features on the mRNA sequence, achieving a Spearman correlation with the amount of protein per mRNA of 0.36 when tested on the dataset of operons' first genes in *E. coli* K-12 MG1655 genome. However, this Spearman correlation was not high, and the model did not fully consider the features of mRNA and protein sequences. Therefore, to enhance the prediction capability, our study firstly tried expanding the testing dataset, adding genes inside the operon, and using the microarray of the mRNA expression data set, thereby helping to improve the correlation of translation rate with the amount of protein with more than 0.42. Next, the applicability of 6 traditional machine learning models to calculate a "new translation rate" was examined using initiation rate and elongation rate as inputs. The result showed that the SVR algorithm had the most correlated new translation rates, with Spearman correlation improving to R = 0.6699 with protein level output and to R = 0.6536 with protein level per mRNA. Finally, the study investigated the degree of improvement when combining more features with the new translation rates. The results showed that the model's predictive ability to produce a protein per mRNA reached R = 0.6660 when using six features, while the correlation of this model's final translation rate to protein level was up to R = 0.6729. This demonstrated the model's capability to predict protein expression of a gene, rather than being limited to predicting expression by an mRNA and showed the model's potential for development into gene expression predicting tools.

**Funding:** The author(s) received no specific funding for this work.

**Competing interests:** The authors have declared that no competing interests exist.

## Introduction

The expression of recombinant protein from the microbial host strains can be considered a major step in microbiological techniques, enabling the high production of protein products for industry and biopharmaceuticals without purifying it from raw materials or using chemical synthesis. The host strain *Escherichia coli* is an extensively used microbial host in protein study and expression due to its strong growth in culture and well-understood genetic information. However, there still has not been an ease-of-use optimization method or tool that could generalize all the predictors affecting the expression from translation initiation to translation elongation (which was the direct protein synthesis step, as well as factors related to post-translational protein stability.

From our understanding, protein expression predicting methods could be divided into methods based on statistical algorithms, such as the ones used in the highly expressed genes prediction tool in the High Expression Gene Database (HEG-DB) [1]. In addition, another expression prediction model introduced by Gilad Shaham and Tamir Tuller could simulate both translation initiation and elongation with the genome used for robustness testing as *E. coli* and integrated into software called Transim. This model can predict the translational dynamics of mRNA based solely on the input nucleotide sequence. However, the model's translation rate did not have a sufficient Spearman's rank correlation with protein level (reported to be up to 0.36), and this could be attributed to the lack of consideration of other translation/post-translation regulation mechanisms [2].

Currently, traditional supervised machine learning methods have been widely used in biological prediction and microbiology to solve predicting and clustering expression levels problems. Particularly, there have been protein expression prediction tools integrating from K-Nearest Neighbor (KNN), Linear Regression and its derivative algorithms, to Random Forest and Support Vector methods (SVM or SVR) [3–6]. However, there seemed to be a lack of a model, statistics-based or machine learning-based, that fully covered mRNA-related features involved in the translation and post-translation steps in *E. coli*. These features could be distributed from the 5'-UTR, like the appearance of an upstream start codon; to the coding sequence (CDS), namely the usage of codon and/or tRNA; or even in the sequential and structural order of protein sequences [7].

The translation was largely regulated at the start of translation in eukaryotes, but this was not entirely true in prokaryotes such as *E. coli*. Here, the rate of large variation between mRNAs mainly depends on the level of codon and tRNA usage [8]. Research on translation dynamics and bottlenecks has emerged from many theoretical studies, such as the TASEP (Totally Asymmetric Exclusion Process) model. In a simple TASEP model, the ribosome binds to mRNA based on an initiation rate, moving randomly with an average initiation rate [9]. The Transim model also adopted this model to simulate translation elongation, with the initiation rate calculated based on the total interaction energy of the 5'-UTR regions. Specifically, the model consists of three calculation steps: (1) calculate the translation initiation rate: as the rate at which the ribosome approaches the start codon, this initiation rate is calculated based on mRNA's interaction with ribosomal RNA; (2) Calculation of translation elongation rate: based on the translation speed of each codon from the ribosomal profiling data to estimate the elongation rate on all codons of a gene; (3) Using the TASEP algorithm with high resolution to simulate the translation using the starting and elongation rates calculated in steps (1) and (2). TASEP provides the ability to predict translation rate, ribosome density, amount of translation termination, and the occurrence of ribosomal jamming. However, the Spearman's coefficient correlation from this model's output with the "true" protein abundance was only 0.36 [2].

Therefore, using this model's translation rate as a starting point, we aimed to further enhance the correlation using machine learning methods and extra features.

The objective of this paper is two-fold; the first objective is applying machine learning methods to calculate a new translation rate parameter (termed "New_TR") from Transim's initiation and elongation rate on different *E. coli* genes, aiming to improve the original "Transim translation rate". Secondly, various features relating to translation efficiency would be assessed and included in the model predicting New_TR with the expectation of further enhancing the correlation between our model's New_TR and experimental protein abundance data. From our knowledge, there has not been a machine learning-based model integrating both translating rate-related features (based on Transim), codon types, and protein stability-related features.

## Material and methods

### Workflow

The process of building a model to predict the amount of protein in *E. coli* is shown in Fig 1. First, mRNA and protein sequence data of *E. coli* K-12 strain MG1655 were acquired and extracted features while being used simultaneously as an input for calculating initiation and elongation rates using Transim. Then, the capabilities of typical machine learning algorithms, including Linear Regression (LR), Ridge, Lasso, Elastic Net, Random Forest (RF), and Support vector regression (SVR), would be examined with *E. coli* initiation rate and elongation rate input and compared with the translation rates calculated by the Transim model. For each algorithm, the mRNA dataset will be divided into training data and testing data with the 80:20 split

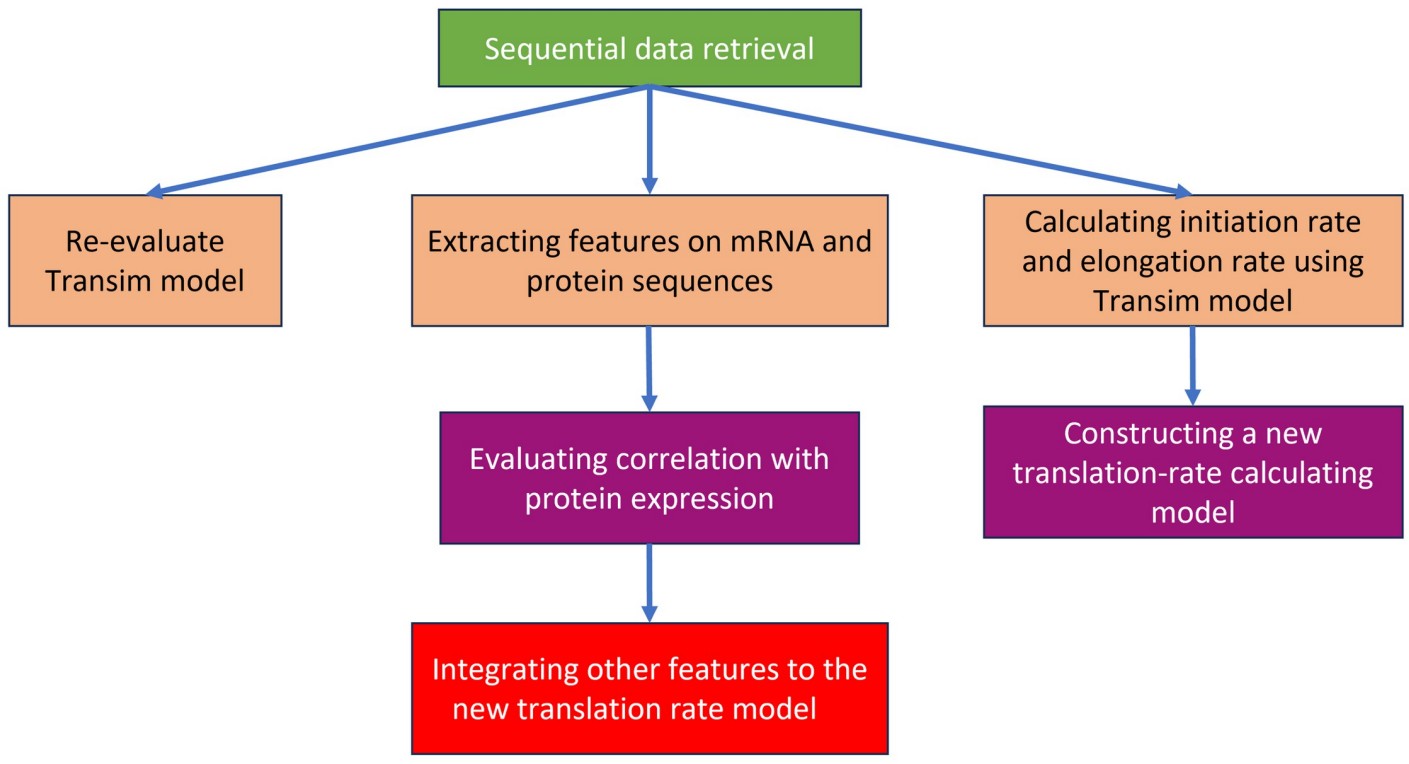

**Fig 1. Model constructing workflow.**

percentage. Finally, the machine learning method calculating the most predictive "new translation rate" will be trained with different combinations of mRNA and protein features combined with the new translation rate, with the expectation of improving the model's predictive power even further.

## Sequential data

4739 CDS sequences of every gene in *E. coli* were retrieved from Gene_sequence.txt from the Regulon database. This is a primary database focusing on *E. coli* transcriptional regulation, which would ensure the validity and specificity of the data compared to general sequence databases such as Genbank [10]. After filtering genes that did not satisfy protein-coding gene requirements (did not have standard start codon ATG, CTG, GTG, and TTG; sequence length was not divisible by 3, or have an internal stop codon), there were 4374 CDS sequences left, which were also translated to protein sequences for further feature extractions.

The 5'-UTR annotations of the first genes were also extracted from the UTR_5_3_sequence file provided by the Regulon database. However, the database's definition of 5'-UTR was just the upstream sequence of an mRNA. Since the translation of different genes can happen simultaneously on a polycistronic mRNA, it is necessary to consider the "in operon 5'-UTR" of genes inside the operons since they might also contribute to translation efficiency. Therefore, gene annotation data from the Ecocyc database was used to retrieve 5'-UTR sequences of "internal genes" and combine them with CDS and protein sequences to extract features needed for subsequent machine-learning steps. 1556 "first gene" 5'-UTR sequences (1) and 1069 "in operon" 5'-UTR sequences (2) were collected from 2 datasets, which were then combined with information on CDS to create 2089 mRNA sequences information with both 5'-UTR and CDS sequences. Ecocyc database was chosen as it is a manually curated biological database dedicated to *E. coli* K-12 that has its data synchronized to RegulonDB. Therefore, it is feasible to combine sequential information from the two databases [11].

## Expression data

Two data sets of protein expression and mRNA expression were used to evaluate the Transim model [2]. First, the protein abundance (PA) level of *E. coli* K-12 MG1655 was obtained from the integrated dataset, the average dataset of all *E. coli* expression data on the PaxDb database downloaded on November 7th, 2020 [11]. Genes with 0 as expression levels were removed from the final composite file. The amount of mRNA expressed in *E. coli* was obtained from the data published previously [12] (referred to as mRNAseq) to replicate the results from the Transim model of Tuller et al. The acquired data was the average amount of mRNA of each gene measured by the RNA-seq method.

In addition, microarray data of mRNA expression from the E COLI EXPRESSION 2 database, which has been acquired and processed since 2019 [13], was also used. Wild-type *E. coli* was grown on an M9 glucose medium (2 g1$^{-1}$) under aerobic and anaerobic conditions for gene profiling experiments, and the measurements were done in triplicates. We processed this data by keeping only the expression profiles of wild-type *E. coli* MG1655 (WT) and removed all the rest containing the mutant strain profiles. For each profile (equivalent to one culture condition), we calculated the mean value of all the replicates [10]. The mean values of all conditions will then be normalized by the quantile normalization method using the preprocessCore library in the R programming language. Finally, we obtain the mean value (mean) of each gene calculated from all microarray experiments, called mRNA expression data from microarray experiment (referred to as mRNA$_{microarray}$).

## Calculating translating rate

The Transim model is a biophysical model that calculates the translation rate based on initiation rate, which embodies the rate of ribosome binding to the start codon, and elongation rate, which is a translation speed of each ribosome simulated by the model's totally asymmetric exclusion process (TASEP). First, initiation rate, elongation rate, and translation rate were calculated by the Transim model, with a text file consisting of names, mRNA sequences (UTR +CDS), and start site positions, using the recommended format of Transim. Sequences with the RBS that were too short for the model to calculate the initiation rate were discarded in the final file. Initiation rate and elongation rate data will then be used as the basis input of machine learning models for calculating a new translation rate with a higher correlation to the expression level compared to the original Transim's translation rate.

## Predicting secondary structures

The secondary structure around the starting codon was able to be predicted using minimum free energy (MFE), which was calculated using the RNAfold program from the Vienna RNA package. The regions under investigation were from the -30 to +30 of the mRNAs, and only the region with less than 15% data loss from insufficient length in the 5'-UTR region would be qualified as a feature for the prediction model.

## Retrieving sequential-related features

The types of start codon and stop codon were analyzed to see their potential as predictive features for the model. The presence of upstream UAG in the 5'-UTR was also assessed, which was classified into three scenarios: in-frame (corresponding to the ORF), out-of-frame, and no uUAG found in the 10-base upstream of the start codon in each mRNA sequence. The codon adaptation index (CAI) and tRNA adaptation index (tAI) were calculated using the webtools CAIcal and stAIcalc, respectively.

Since prokaryote's signature protein synthesis mode is polycistronic translation, the effect of gene positions in an operon was considered. Particularly, in this model, we focus on whether it was the first gene of an operon or not.

## Retrieving protein stability-related features

The stability *in vitro* of the protein was quantified by the Instability index, which was calculated by the *instaIndex* package in R language. Meanwhile, the half-life of proteins was based on the N-terminal rule, in which the half-life of proteins in *E. coli* cells would be labeled either "under 2 minutes" or "over 10 hours" according to the type of residue at their N-end (Proline N-terminal proteins were not included due to not being able to be determined based on N-terminal rule alone).

## Machine learning model

Six traditional machine learning models, namely LR, Ridge, Lasso, Elastic Net, Random Forest (RF), and Support Vector Regression (SVR) are analyzed to choose the most suitable for estimating protein expression. The Scikit-learn library provided these algorithms on the Python platform [14]. The input dataset has been labeled and split into the training and testing sets with the 80:20 ratio. Machine learning methods can be then used to train the model based on the training set inputs and the labels of these inputs, then find the relationship between the input and the label.

### Evaluate features and model's output

In order to represent features or outputs in real, continuous values, Spearman's Rank correlation coefficient was used to evaluate their correlation with protein abundance (PA) or protein abundance produced by one mRNA (PA/mRNA). Such a measure was chosen for its simplicity in usage and effectiveness in assessing monotonic relations. Also, using the same metric as the one used in Tuller's report would give us a clear comparison of the performance of the two paper's methods.

For categorical features like uAUG or types of start and stop codons, we used Student's t-test to determine the two-tailed significance of the difference between groups of genes. The statistical analysis is conducted by the Scipy library running on a Python environment.

## Results

### Re-evaluate the Transim model

Our first objective was to evaluate whether the calculated translation rate from Transim was like the value previously reported. Thus, we calculated the translation rate of the first genes, similar to the data set used to evaluate the Transim model. The correlation test of translation rate showed the Spearman coefficient with protein expression per mRNA was R = 0.3810, equivalent to the result in the paper of Tuller *et al.*, which was 0.3593 (**S1 Fig**). Thus, it was safe to assume that our research has replicated the results from the Transim model published in the article.

Since the RegulonDB dataset used to evaluate the Transim model only included the first gene of each operon, we sought to improve the original dataset as there are still many genes inside the operon that might possess valuable information for the model. The subsequent genes in the operon were thus collected of their mRNA sequences and determined if the 5'-UTR sequence region of the genes inside the operon had the same properties as the first genes data from RegulonDB. Then, Nucleotide usage in the -15 to +15 region was analyzed. **Fig 2** shows the most frequent nucleotide in the 5'-UTR region of the first genes in the operon (A) and the subsequent genes (B) as a logo sequence using the web logo tool. The results showed that, in both cases, an A/G-rich region appeared six nucleotides away from the start codon, corresponding to the Shine-Dalgarno sequence. Specifically, in **S1 Table**, we could see that the regions -12 to -7 of both groups had over 30% chances of Guanin (G) occurrence, especially in the two positions -8 and -7 of both groups where G occurrences were more than 50%. This suggested that the Shine-Dalgarno (SD) motif had the same sequences in both groups. In addition, when looking at 30 nucleotides around the start codon, we found a relatively high rate of Adenine usage, consistent with the previous observation that the A-rich region around the start codon was a signal for the ribosomal subunits to enter transcription initiation, especially in the absence of a Shine-Dalgarno sequence or a weak motif [15]. Thus, it could be concluded that the first genes and the genes inside the operon had similar translation initiation mechanisms, and this additional data could be used to train and evaluate the efficient prediction model as the result of the encoding process.

At the same time, when evaluating the correlation of translation rates, the Spearman correlation of these sequences was R = 0.3546 (only a 3% decrease compared to the correlation considered with the first gene). Still, the dataset was 30% more abundant than the original dataset (**Fig 3A**) [2]. The slight decrease in the correlation coefficient might be attributed to numerous factors other than those considered in the Transim TASEP model arising when extra data were included. All in all, this combined dataset would be appropriately used for later analyses and model training.

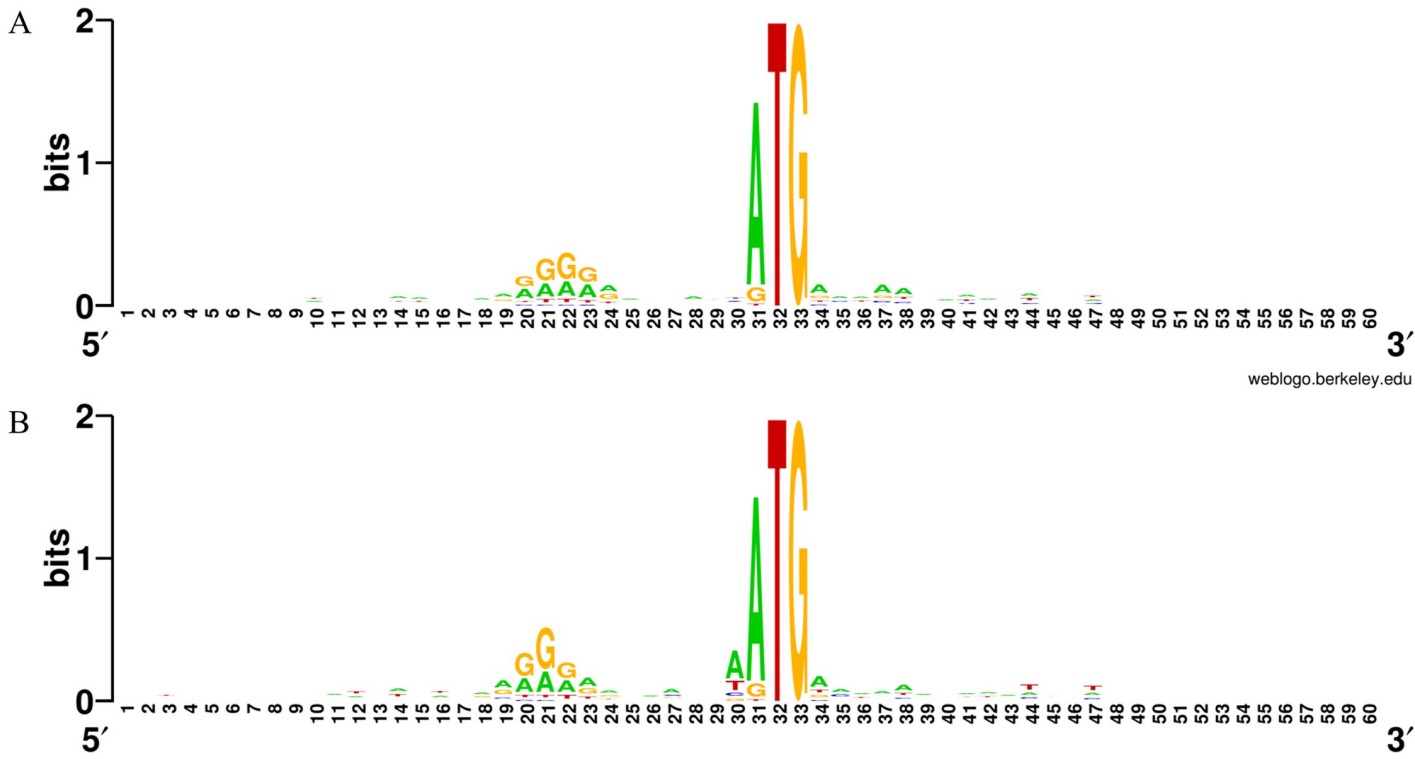

**Fig 2.** Sequence logo of the region from -30 to +30 in (A) first genes and (B) genes in the operon.

However, an unexpected result was observed when the output was PA instead of PA/ mRNAseq. This time, the correlation of the translation rate increases by more than 20% and reaches R = 0.5623 (**Fig 3B**), although the distribution on the histogram was still similar. Thus, using PA/mRNAseq expression data could have reduced the correlation coefficient. In other words, the RNA-seq data used by Tuller might have affected the correlation of the Transim model. We addressed this issue by using microarray data obtained from E COLI

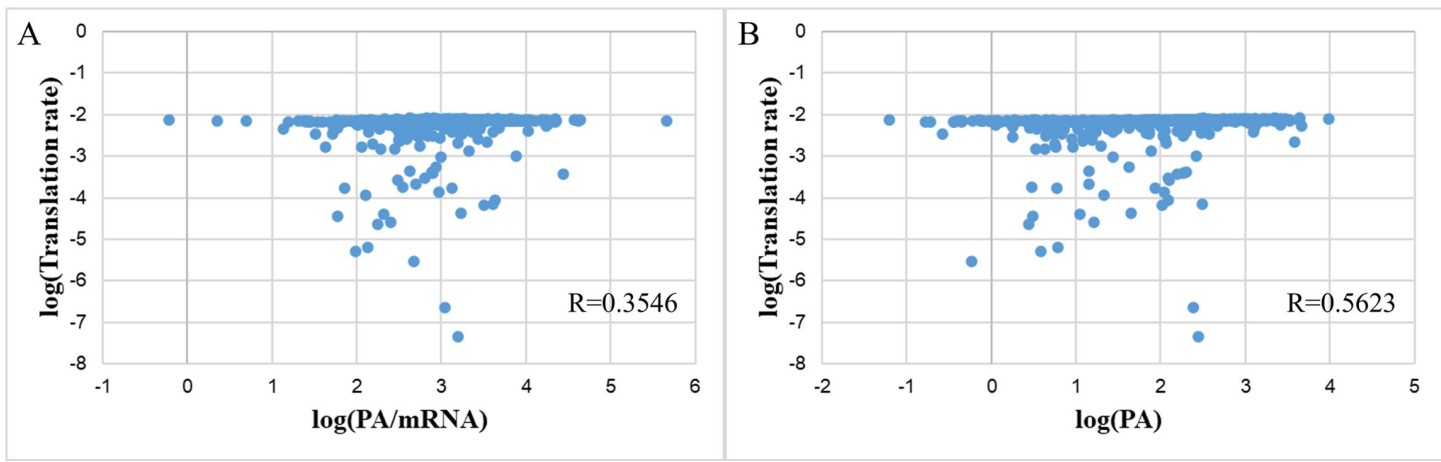

**Fig 3. Correlation between first genes' PA/mRNA and their translation rates calculated by the Transim model.**

EXPRESSION 2 instead of mRNAseq data. The experimental results showed a significant improvement in the correlation, reaching 0.4316 with even more sequence data with available mRNA levels (Fig 4A). Moreover, the correlation was maintained at around R = 0.43 when the output was protein abundance (Fig 4B). It shows that the microarray value was more suitable and consistent than the RNA-seq data used to calculate the correlation. One possible explanation was that one could improve the accuracy of the mRNA expression data by using data averaged from multiple experiments under different conditions rather than just one publication. In other words, PA/mRNA with mRNA expression from microarray experiments is used as an evaluating metric for later analyses.

## Effect of the position of genes in operon on the expression level

Even though Spearman's rank coefficient improved with mRNA microarray output, the correlation was still not satisfactory, which led us to extract more mRNA and protein sequential features that could contribute to different translation regulation steps. In the end, a total of 6 groups of features were collected, namely: the position of genes in the operon (first gene or in-operon gene), 5'-UTR feature (upstream AUG—uAUG), CDS (start codon, stop codon, CAI, and tAI), folding energy (+1 to +30 region), protein-sequence (N-terminal half-life, Instability index, length), and translating rate (initiation rate and elongation rate calculated by Transim model).

First, the influence of the gene's position in the operon on the PA/mRNA level was measured (Fig 5). Despite showing no difference in the 5'-UTR nucleotide frequency between the first gene and in-operon gene categories, when considering the expression level of the two groups, there was a stark difference between the two groups, with the first-gene group being more highly expressed (P = 1,7x10$^{-3}$). Even though the result has not been shown in other models like Transim, the superiority of the first genes could be attributed to the SD motif strength and the ribosome competition between the first and subsequent genes.

## Effect of 5'-UTR's upstream AUG features on protein level

Factors affecting expressions in the 5'-UTR region, such as the folding energy or the interaction between the SD region and the ribosomal subunit, were already included in the folding energy and translation initiation rate feature groups, so we only considered a type of sequential

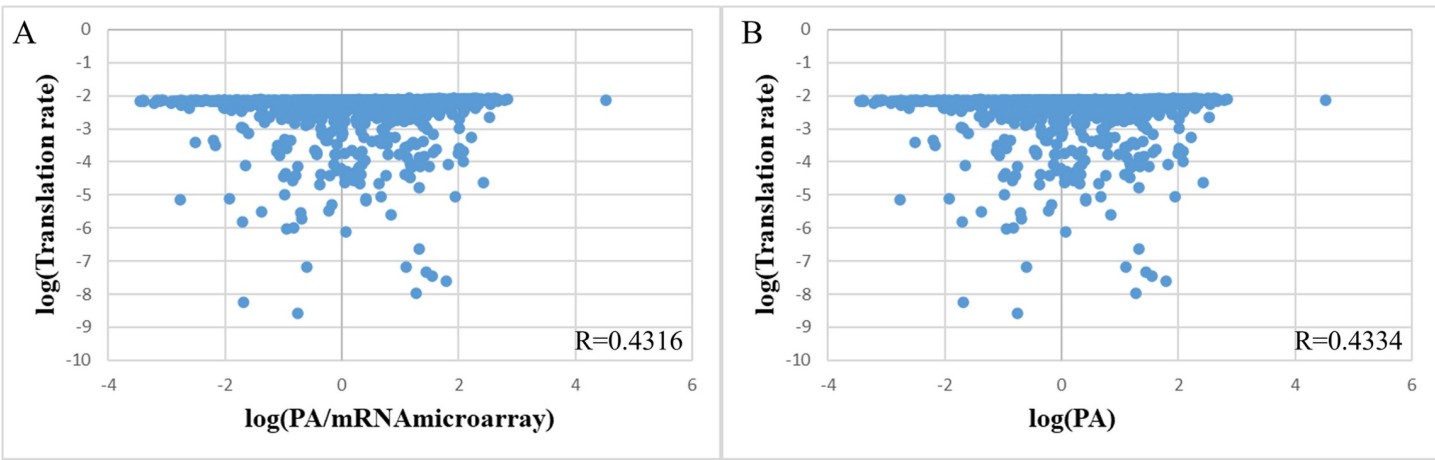

**Fig 4.** The correlation between translation rates and (A) PA/mRNA with mRNA expression level from normalized microarray data and (B) PA.

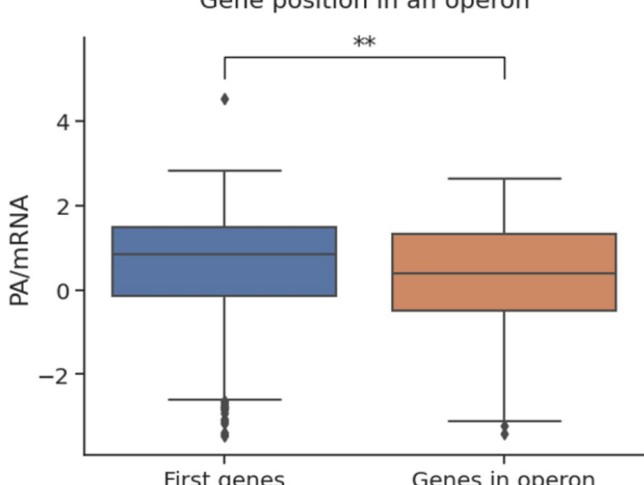

**Fig 5. Protein abundance per mRNA for first genes and subsequent genes in the operon.**

motif that might affect the expression here—the AUG-type start codon in the upstream region (uAUG). Based on previous research on the impact of the expression of uAUG on yeast, the study only investigated the occurrence of uAUG in the region from -12 to +1. Out of all the examined genes, 90.9% did not possess uAUG motifs, while 2.8% of genes existed with an inframe uAUG, and the rest were genes with uAUG out of frame with the correct reading frame of the gene.

The presence of uAUG has been shown to affect 6% of the yeast protein output if the uAUGs were in a frame different from the open reading frame. However, studies on uAUG are still focused on eukaryotes such as yeast [16]. Based on these results, the study was conducted to see whether this motif had the same effect on *E. coli*. It was shown that the presence of frameshift uAUGs reduced PA/mRNA significantly (P = 0.044). In contrast, there was no significant effect of the in-frame uAUG on protein expression (P = 0.4340) (**Fig 6**). This was explained by the fact that the out-of-frame codons would trigger the wrong and degraded

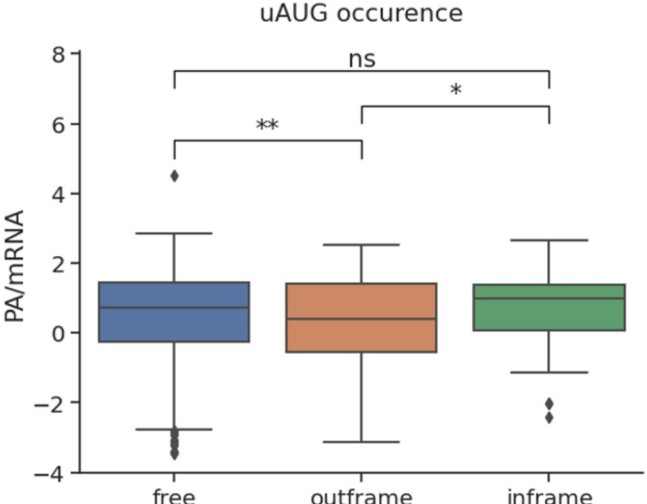

**Fig 6. Differences in the expression level of three gene groups with different types of uAUG motif.**

protein products, while the genes with in-frame uAUG could still produce the target sequence with a longer fragment located upstream. This suggested the existence of a regulatory mechanism upstream of *E. coli* initiation codon expression similar to that found in the yeast *S. cerevisiae* [7].

### Effect of coding sequences features on protein expression level

In addition to the commonly used codon ATG, *E. coli* also uses TTG and GTG as synonymous codons as initiation signals, affecting ribosomal binding and translation initiation efficiency [17]. ATG is the most favored initiation codon in *E. coli*, accounting for 89.8% of the 1660 genes used. Therefore, it is understandable that using the start codon ATG gave a much higher amount of protein than using the next two common codons, GTG (P = 0.014) and TTG (P = 0.040). Meanwhile, there was no apparent difference when the start codon was GTG or TTG (P = 0.2800) (**Fig 7A**).

Similarly, when considering the effect of the stop codon types on expression, we also see the emergence of a codon that was more dominant and was also the codon with the highest expression rate. TAA was the choice for the stop codon of 69.70% of genes, and the group

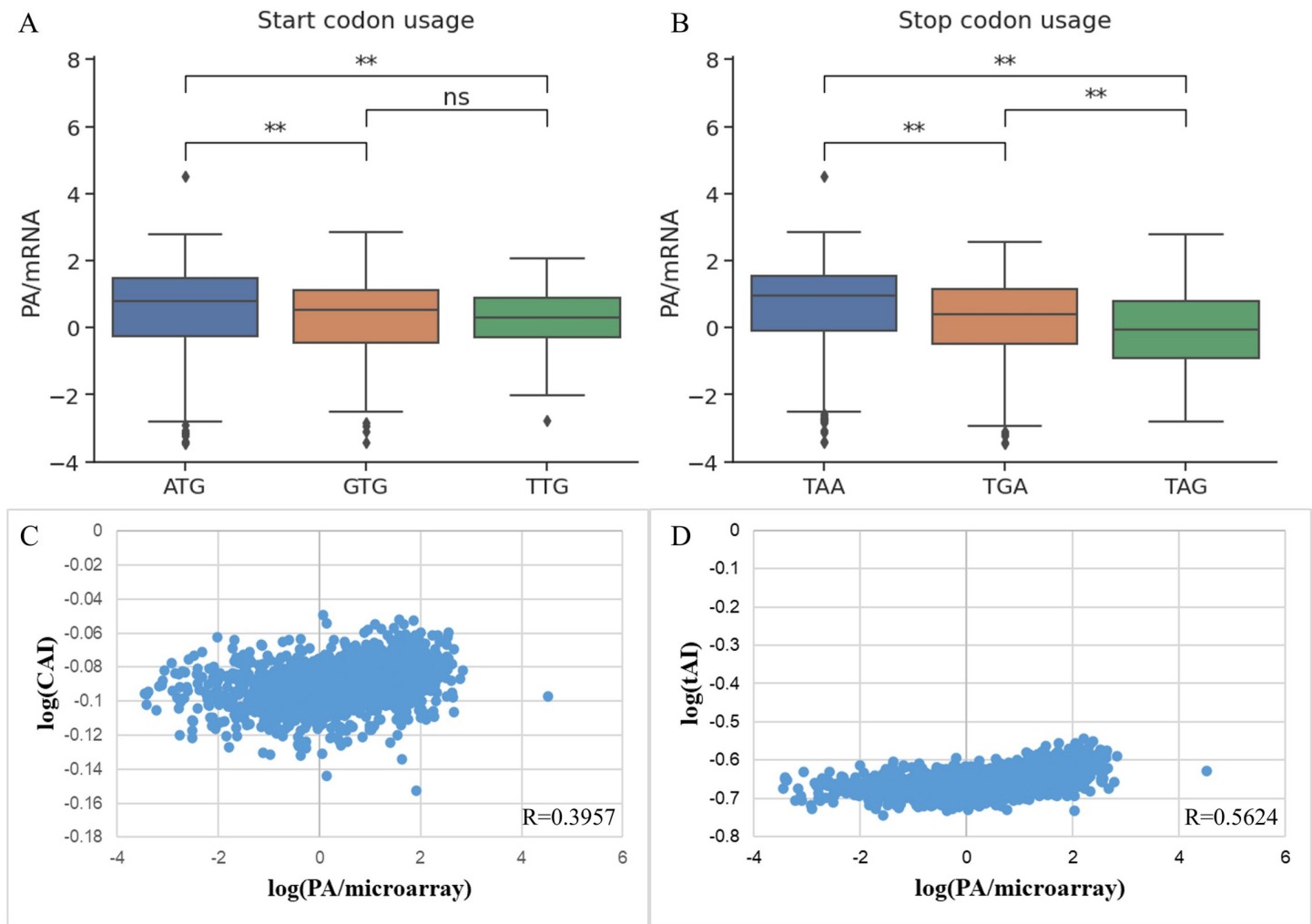

**Fig 7. Effect of features on the coding sequence to the protein expression level per mRNA.**

using this codon had a much higher protein production efficiency than TGA and TAG ($P \ll 10^{-4}$). In addition, there was a difference between the other two codons, with the group using the TGA codon giving a higher protein output than TAG (P = 0.01) (**Fig 7B**). This was a new result, as there were currently no published applications of end codons in protein expression prediction models.

Codon and tRNA usage were two quantifiers that characterize codon bias in genes and have been associated with elongation rates. Therefore, **Fig 7C and 7D** showed similar distribution trends in CAI and tAI, which were highly positively correlated with protein per mRNA, reaching Spearman R's coefficient of 0.3957 and 0.5624 on the total dataset, respectively. It suggests that features related to the rate of translation elongation have a more significant impact on protein output than factors related to translation initiation. In contrast, a previous report claimed initiation of translation was the rate-limiting step [8]. However, the correlation with protein levels of the elongation rate calculated by the Transim model supported the correlation results of CAI and tAI with protein/mRNA [2].

## Effect of secondary structure on protein levels

Finally, the study considered the effect of the secondary structure around the start codon. After the process of calculation and statistics, our research would only retain the folding energy of the region +1 to +30 since the other assessed regions missed from 17% to 33% of the data to insufficient lengths. However, this would not significantly affect the model strength since the folding energy of the region -35 to +35 was already included in the calculation of the initiation rate. Surprisingly, the folding energy of this region had almost no impact on the amount of protein per mRNA (R = 0.0039) (**Fig 8**). However, this was consistent with the previous observation that the initiation rate factor had no significant effect on protein output (R = 0.1350), which was calculated as the total energy of conformation and was equivalent to the 5'-UTR region of the mRNA. In addition, when considering the correlations of other folded regions, such as -30 to +30, -30 to +1, or -15 to +15, the results also did not show a significant effect on protein output (R <0.15; **S2 Fig**). This result was quite contrary to the observations of strong effects on translation initiation and translation elongation of mRNA, as well as the existence of a logistic regression model predicting the likelihood of expression only based on features related to the secondary structure [6]. However, it can be explained that the folding energy characteristics, if considered individually and not in correlation with the SD sequence region, will not be highly correlated because the presence of strong SD sequences helped overcome structural barriers [18, 19]. At the same time, the combination of folding factors had been realized through the Initiation Rate calculated by the Transim model, and introducing too many overlapping features could cause the model to be overfitting.

## Effect of protein sequence-related features on protein level

First, the effect of protein or mRNA length was minimal on the protein expression (R = 0.0867) (**Fig 9A**). Meanwhile, the N-terminus on post-translational protein degradation was compared between the two groups: the group that was predicted to have a half-life of more than 10 hours and the group predicted to have a short half-life of fewer than 2 minutes. Proline N-terminals have been excluded because there was no clear data on which group this tag belongs to. There was a statistically significant difference between the two groups, in which the protein group with a half-life of more than 10 hours had more total protein ($P = 3.0 \times 10^{-4}$) (**Fig 9B**). Finally, with the protein instability index II (Instability index), there was a negative correlation between this index and the expression level (Spearman R = -0.1655 and -0.1988 on the test set) (**Fig 9C**). This was consistent with the definition of II, where if this index exceeds 40,

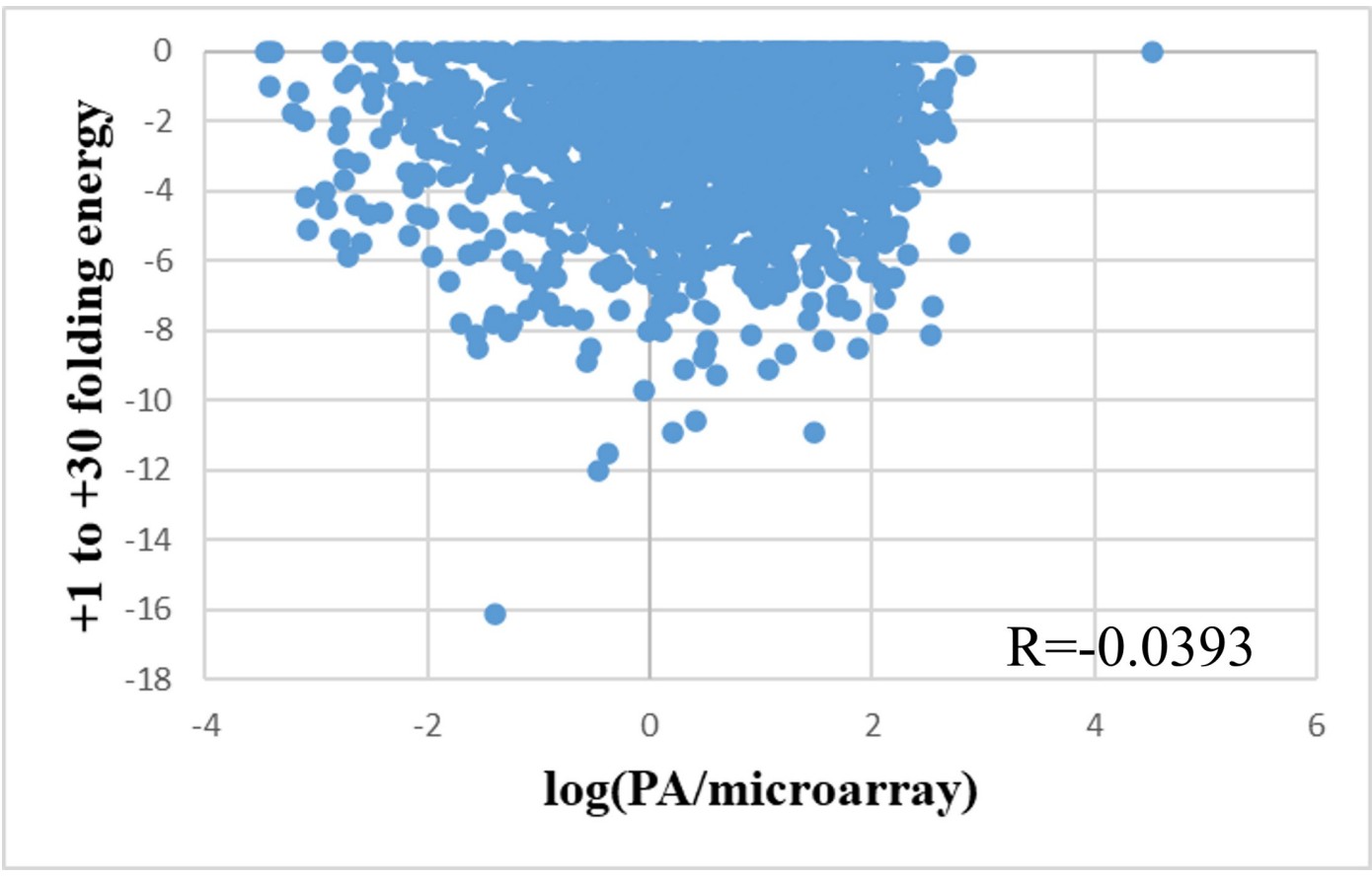

**Fig 8. The correlation between PA/mRNA and the folding energy in the +1 to +30 region of the mRNAs.**

the protein is considered unstable. However, all three predictors would still be included in the final model as their combination might affect the predictive capability differently.

### Effect of translating rate-related features

Our research examined the correlation between the individual components of the translation rate, namely the initiation rate and elongation rate, and the protein expression per mRNA in **Fig 10**. While the initiation rates were weakly correlated with PA/microarray (R = 0.1350 and had a distribution comprising of two groups (0.01 to 1 s$^{-1}$ and $1.0 \times 10^{-8}$ to $1.0 \times 10^{-3}$) (**Fig 10A**), elongation rates gave a near-linear correlation with PA/microarray but with a narrower range of rate distributions (maximum elongation rate was only 1.26 times higher than the minimum values—**Fig 10B**). In addition, a much higher Spearman correlation was on the whole dataset to that of the translation rate (0.6538 to 0.4268). Hence, the combination of initiation and elongation rates using Transim's TASEP-based model was still not optimized and was heavily influenced by the initiation rate.

### Formulating a new "Translation rate" using machine learning methods

In an attempt to increase the correlation between the translation rate and protein per mRNA level, we first try to create a "new translation rate" (New_TR) metric by six traditional

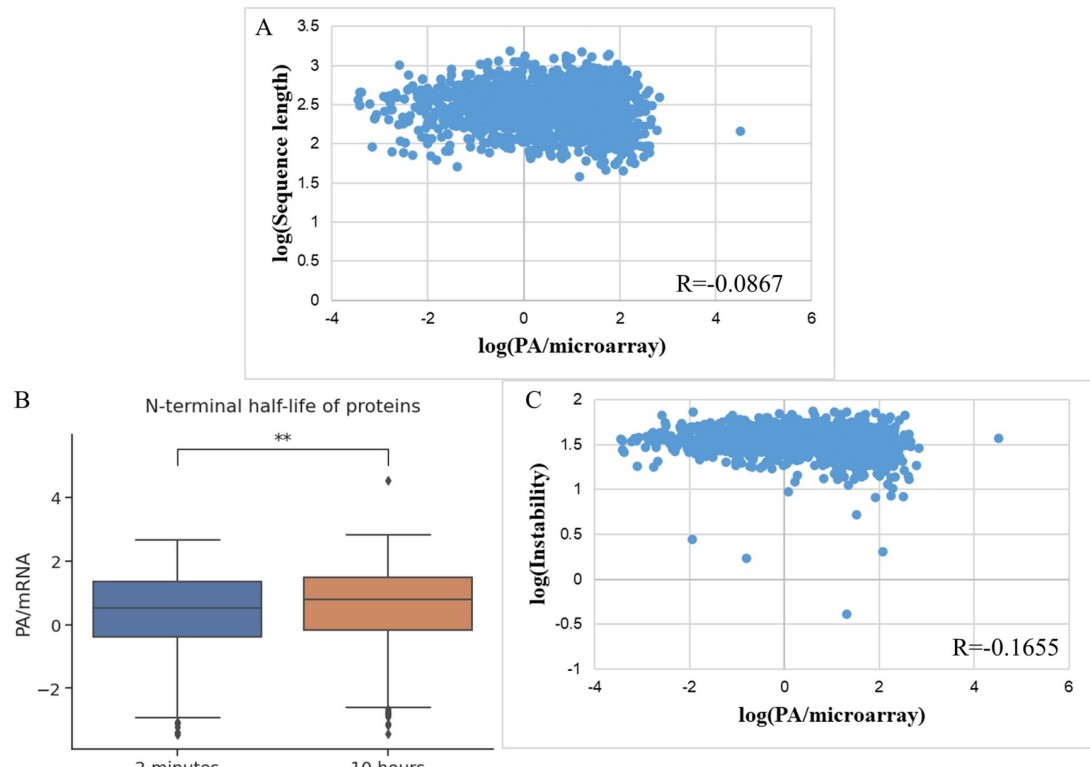

**Fig 9.** Effect of A. Protein sequence length, B. Protein half-life based on N-terminal rules, and C. The instability index (II) of protein.

supervised machine learning methods with the same initiation rate and elongation rate input as the Transim one to predict PA and PA/mRNA.

First, 1660 input samples were divided into two training and test sets with a ratio of 8:2. Then, the Spearman correlation between New_TR of genes with PA and PA/mRNA is presented in **Table 1**, where higher values are highlighted in red, and lower values are marked in blue. We, therefore, have observed that when evaluated on the test set, New_TR from the

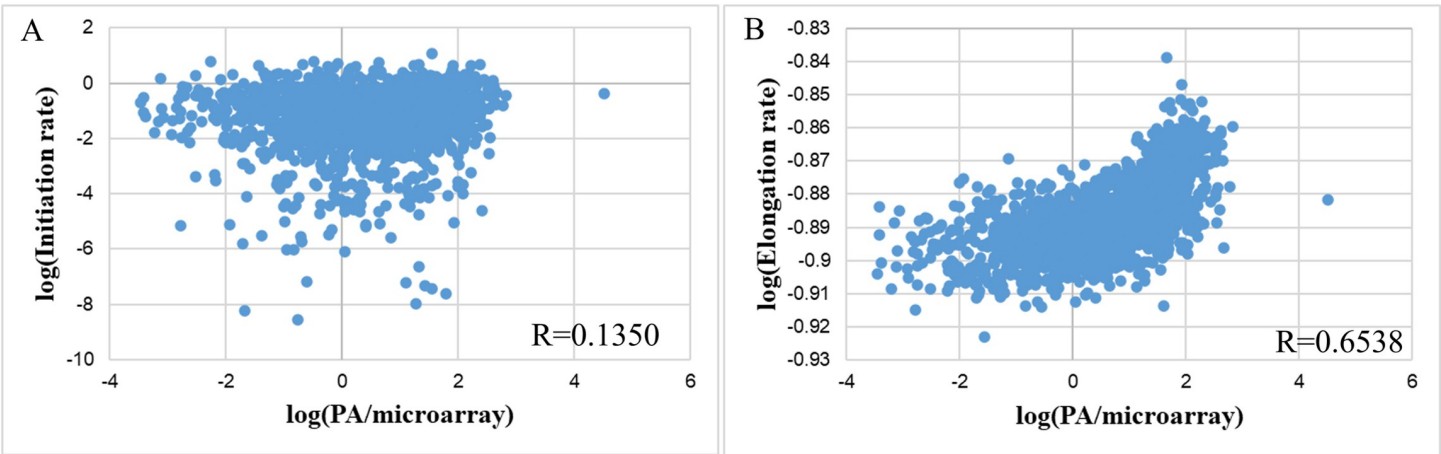

**Fig 10.** The correlation between PA/mRNA and (A) initiation rates and (B) elongation rates.

**Table 1.** Correlation between new translation rate calculated by different machine learning models and PA/mRNA or PA on A. Testing dataset and B. Total dataset.

| | LR | Lasso | Ridge | Elastic Net | SVR | RF |
|---|---|---|---|---|---|---|
| **A. Testing set** | | | | | | |
| PA/mRNA | 0.6191 | 0.6194 | 0.6191 | 0.6187 | 0.6178 | 0.5701 |
| PA | 0.6313 | 0.6323 | 0.6313 | 0.6305 | 0.6342 | 0.5608 |
| **B. Total dataset** | | | | | | |
| PA/mRNA | 0.6503 | 0.6513 | 0.6503 | 0.6499 | 0.6536 | 0.5694 |
| PA | 0.6650 | 0.6652 | 0.6650 | 0.6640 | 0.6699 | 0.5232 |

Lasso algorithm gave the highest correlation with PA/mRNA, reaching R = 0.6194, and with output as PA, New_TR from SVR had the highest Spearman coefficient, reaching R = 0.6342. However, when evaluating the Spearman coefficient on all 1660 data, New_TR from SVR showed the highest correlation in both PA and PA/mRNA cases (**Table 1**). Therefore, the SVR machine learning algorithm was chosen as a basis to build the expression prediction model because this algorithm gave the highest correlation in translation rate in 75% of the scenarios. At the same time, it was clear that even the most poorly performed RF had a Spearman coefficient of 20% to 30% higher than the translation rate from the Transim model. Meanwhile, new_TR from SVR improved correlation from 51.4% (0.4316 to 0.6536) with PA/mRNA output to more than 54.6% with PA output (0.4334 to 0.6699), showing the capability of machine learning models, especially SVR, in tackling protein expression prediction problem.

The computed results showed that the "new translation rate" improved the prediction of protein expression to R = 0.6536 and R = 0.6699, with the output being PA/mRNA and PA, respectively (**Fig 11A and 11B**). The results also illustrate that the translation rate value could now be assigned with negative values. It was because the machine learning model will predict the new values that best fit the equation that was built from the training set. At the same time, it was found that in both the protein and protein/microarray cases, translation rates tend to be positively correlated with the amount of protein or the amount of protein produced by an mRNA, reinforcing the predictive power of this new model, when comparing to previous mathematical models on the correlation of translation rate with protein amount [20].

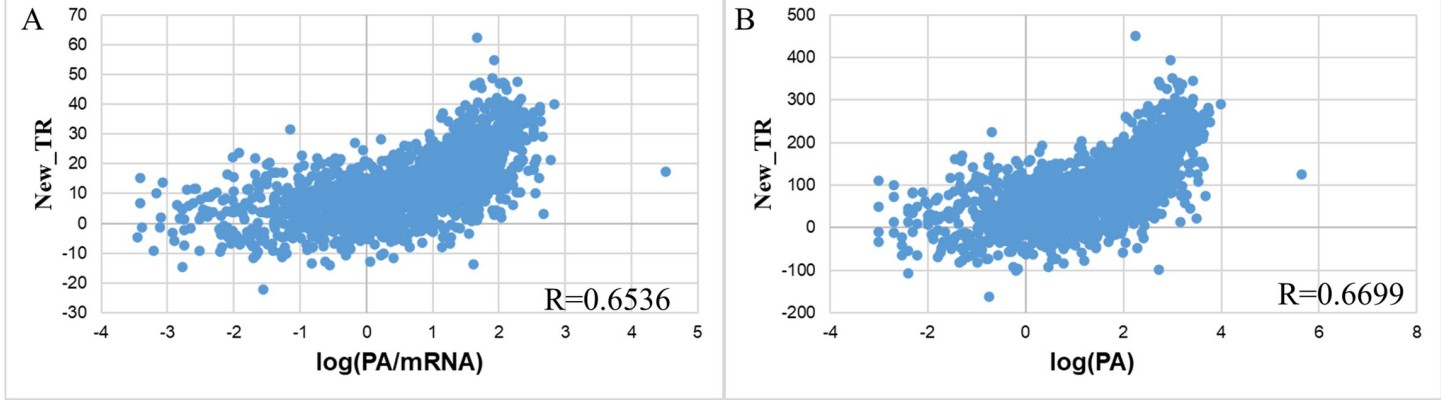

**Fig 11.** The improved correlation between "New translation rate" and (A) protein expression and (B) protein expression per mRNA.

## Analysis of the best feature combinations for SVR

From the analysis in the previous section, 11 sequential features were used to train the SVR model, including new translation rate (New_TR), positions of the gene at the beginning or in the operon (Type); the appearance and relative position of the codon AUG before the open reading template (uAUG); codon usage (CAI) and tRNA usage (tAI); Type of start codon (Start) and Type of end codon (Stop); sequence length (Length); the half-life of the protein was based on the N-terminal (N-terminal) and the protein instability index (Instability); and the folding energy ranges from +1 to +30 (+1 to +30).

New_TR is the original feature to combine with other features because this was a feature that could generalize both the initiation rate and the elongation rate at the translation level, and the goal of our research was to improve the translation rate from the Transim model.

After training the model with New_TR and each remaining feature separately, we evaluated the Spearman correlation of the prediction results on the test set with the outputs PA/mRNA and PA. **Fig 12** showed that the correlation of the model was not significantly improved when combining single factors and even tended to decrease with most features. Only Start, Stop, and Type were able to improve the PA/mRNA prediction when combined with New_TR, whereas the only three features that raised the correlation with the PA output were CAI, Length, and Type. Furthermore, the best feature to enhance the model's performance in both outputs was Type, reaching R = 0.6224 with PA/mRNA and R = 0.6388 with PA. With the features not likely to improve New_TR, the combination of the CAI feature for predicting PA/mRNA and tAI for predicting PA reduces the correlation of the model (by 1% compared to the original New_TR). However, these were two features highly correlated with the amount of protein expressed (RCAI-PA/mRNA = 0.3937 and RtAI-PA/mRNA = 0.5637). One possible reason is that when calculating the elongation rate using the TASEP model, the sliding time per codon (Typical decoding rate) was of the same quality as tAI or CAI, which caused the model to be overfitted. Although there has been a significant improvement in the model when combined with the location feature of the gene in the operon, the question was whether combining with the specific combination would somewhat improve the model's correlation. Therefore, our subsequent research will investigate all possible combinations of features to include in the new translation rate model.

## Analysis of different feature combinations to the SVR model

The study combined 2 to 10 features for the new translation rate prediction model, and the best-performed combination to predict PA and PA/mRNA is presented in **Fig 13**. One can see that the model had the best predictive ability when combining features from 3 to 7, and the predictive power starts to decrease when combining eight or more attributes. It may result from overtraining, where the model tried to find an equation to account for too many features in the training set. It may contain identical information leading to the inability to generalize and not apply well to actual data.

With the output as PA/mRNA, combining six features: folding energy of +1 to +30 region, N-terminal Half-life of protein, Start, Stop, Length, and Type gave the best correlation, reaching R = 0.6260 and improving the model's prediction by 0.82% (**Fig 13A**). As for the output of total protein, the combination of 4 features, Start, Stop, Length, and Type, gave the best predictive results of translation speed, reaching 0.6418 and increasing by 0.76% compared to the New_TR (**Fig 13B**). In addition to the position of the gene in the operon, which is always a factor that helps to improve the best model when examining the combination of 1 to 10 features, the three features Start, Stop, and Length also appeared in 80% of the best combinations (**S2 Table**). It is

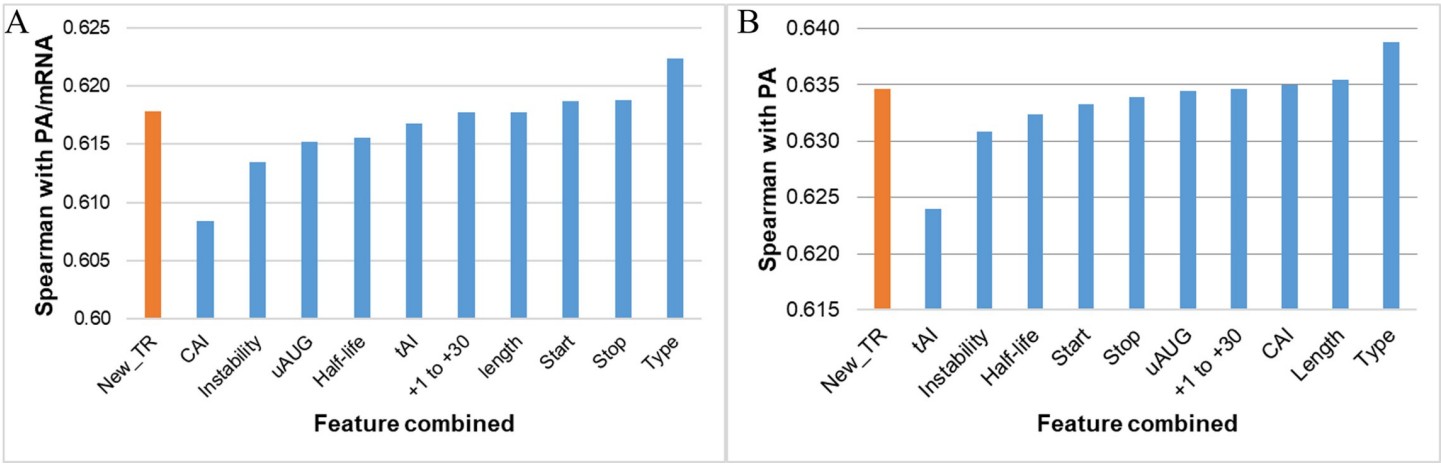

**Fig 12.** Performance of SVR when predicting (A) protein expression and (B) protein expression per mRNA while integrating every single feature.

a new result because sequence length had a very low correlation with protein expression (R = 0.08); no model has yet incorporated a stop codon as a prediction feature.

Then, to compare with the Transim model, the study used two models for the best correlation on the test set for two outputs, PA/mRNA and PA, to predict all 1660 samples. The results showed that the Spearman coefficient of both cases increased sharply compared to the results on the test set, reaching a correlation of RPA/mRNA = 0.6660 and RPA = 0.6729, respectively. The model, however, only improved by 1.9% with PA/mRNA output and 0.5% with PA output (RPA/mRNA increased by 0.0124 and RPA increased by 0.0030) when incorporating more features compared with the original New_TR. It is still about 55% better than the 2018 study by Tuller et al. (RPA/mRNA increased by 0.2344 and RPA increased by 0.2398), although also based on the input of initiation rate. The initiation and elongation rates were calculated using the Transim model. At the same time, the influence of codon choices, the sequence length, and especially the gene's position in the operon (which appears in all the best combinations) on translation levels were highlighted in this model.

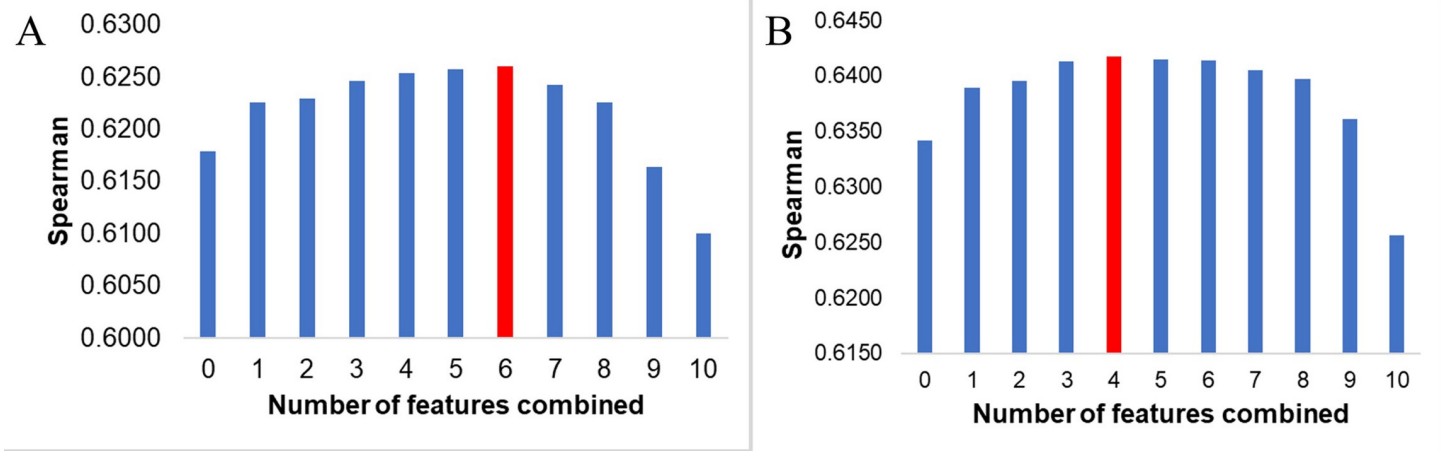

**Fig 13.** The best Spearman correlation when combining from 0 to 10 sequential features to the model to predict A. PA/mRNA and B. PA.

## Discussion

In order to build a model capable of predicting the protein expression of *E. coli* based on the input mRNA sequence only, the study relied on the current translation rates calculated from the Transim model to build a more well-rounded translation rate model. First, the protein expression and mRNA expression data were extracted to evaluate the model, and the mRNA sequence was used to calculate the translation rate using the Transim and feature extraction to build a new predictive model. In addition to the first genes of operons in the Regulon dataset, gene sequences in the operon were also included in our mRNA sequence data to diversify the data while also considering the effect of gene positions on expression level. Our research has repeated the results from Transim's publication, with the Spearman correlation between the translation rate and PA/mRNA reaching 0.3810, equivalent to the published Transim correlation results of the first genes dataset (R = 0.36) [2]. In addition, adding mRNA sequence data of genes in the operon did not significantly reduce the correlation (R = 0.36). As the correlation of translation rate increased dramatically to R = 0.5623 when removing the mRNA expression from the mRNA-seq data of the article, our research decided to use the mRNA data from the experiments. It improved the correlation of translation rate with PA/mRNA to R = 0.4316. At the same time, this value does not decrease when using the output as PA, indicating the reliability of the data.

Next, the mRNA sequences can be used to obtain 13 features, including the position of the gene at the beginning or in the operon; the appearance and relative position of the codon AUG before the open reading template (uAUG); CAI and tAI indexes; type of start codon and type of stop codon; sequence length; half-life of protein that based on N-terminal and protein instability index, folding energy region from +1 to +30 (+1 to +30), initiation rate, elongation rate, and translation rate from Transim. Then, our research evaluated the correlation of each of these features with the amount of PA/mRNA and found that the correlation of translation rate from Transim had a lower Spearman coefficient (R = 0.4316) than that of the elongation rate (R = 0.6538), even if it was a combination of the initiation rate (R = 0.1350) and the elongation rate. Therefore, our research has calculated the new translation rate (New_TR) using six supervised machine learning models with the same input as the initiation and elongation rates calculated by the Transim model. As a result, with output PA and PA/mRNA and considering the test set, Lasso was the best model for correlation with PA/mRNA (RPA/mRNA = 0.6194), and SVR gave the best correlation with PA (R = 0.6392). When considering the whole mRNA sequence, SVR gave the best predictive ability with both PA and PA/mRNA outputs, reaching Spearman correlation coefficients of RPA = 0.6699 and RPA/mRNA = 0.6536, respectively. Since it showed the ability to give New_TR with better correlation than the remaining models in 3/4 of the cases, SVR was considered the appropriate algorithm to build the expression prediction model. In addition, the original goal of using the amount of protein expressed per mRNA as an evaluation criterion was that the authors who built the Transim model wanted to negate the impact of mRNA-dependent regulatory factors [2]. Therefore, one could explain the higher PA relative to PA/mRNA correlation by including one or more mRNA-dependent features in the initiation and elongation rate characteristics.

In summary, our model improved the accuracy in predicting gene expression in *E. coli* K-12 MG1655 compared with the previously published Transim model using the SVR machine learning algorithm and integrated other features on mRNA and protein sequences besides initiation and elongation rates. Despite this, since this research was used as a base for further development of expression prediction tools, the data used for training might be able to expand further. Furthermore, we aim to study the model further to develop it into a convenient prediction tool for research purposes.

## Supporting information

**S1 Fig. The correlation of Transim's translation rate to the PA/mRNA.** The dataset used here was based on the publication paper of the Transim model (Tuller et al., 2018).
(TIF)

**S2 Fig. The correlation between PA/mRNA and the folding energy.** From A to D, the Spearman's rank coefficient between PA/mRNA and folding energies in the -30 to +30, -15 to +15, -30 to +1, and +30 to +1 regions of the mRNAs. The sequential data here were the same for each analysis, which was 33% less than the total dataset, and hence the difference in Spearman's coefficient of +1 to +30 folding to PA/mRNA level.
(TIF)

**S1 Table. The nucleotide frequency at each position in the -15 to +15 region.** The first column on the left showed the nucleotide frequency for the first genes in the operons, while the second column on the right showed the frequency from subsequent genes in the operons. The putative Shine-Dalgarno regions were noted in bold, while the color indicated occurrence percentage, with red cells being more used nucleotides and blue cells being less commonly used nucleotides.
(DOCX)

**S2 Table. The Spearman coefficient of every feature combination to the new translation rate model.**
(DOCX)

**S1 File.**
(RAR)

## Acknowledgments

The authors thank AISIA Research Lab for the assistance of machine learning model-building.

## Author Contributions

**Conceptualization:** Nam T. Vo, Hoang D. Nguyen.

**Data curation:** Nhat H.M. Truong, Nam T. Vo, Binh T. Nguyen, Hoang D. Nguyen.

**Formal analysis:** Nhat H.M. Truong, Binh T. Nguyen, Son T. Huynh, Hoang D. Nguyen.

**Investigation:** Nhat H.M. Truong, Binh T. Nguyen, Son T. Huynh.

**Methodology:** Nam T. Vo, Binh T. Nguyen, Son T. Huynh, Hoang D. Nguyen.

**Project administration:** Nam T. Vo, Binh T. Nguyen.

**Resources:** Binh T. Nguyen, Hoang D. Nguyen.

**Software:** Nhat H.M. Truong, Nam T. Vo, Binh T. Nguyen, Son T. Huynh.

**Supervision:** Nam T. Vo, Binh T. Nguyen, Hoang D. Nguyen.

**Validation:** Nhat H.M. Truong, Nam T. Vo, Binh T. Nguyen, Son T. Huynh, Hoang D. Nguyen.

**Visualization:** Nhat H.M. Truong, Nam T. Vo.

**Writing – original draft:** Nhat H.M. Truong.

**Writing – review & editing:** Nhat H.M. Truong, Nam T. Vo, Binh T. Nguyen, Son T. Huynh, Hoang D. Nguyen.

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
