## [Decision Letter · Decision Letter 0]

28 Apr 2023

PONE-D-23-08391Analyzing the correlation between protein expression and sequence-related features of mRNA and protein in Escherichia coli K-12 MG1655 model

Correlation between protein expression and sequential featuresPLOS ONE

Dear Dr. Nguyen,

Thank you for submitting your manuscript to PLOS ONE. After careful consideration, we feel that it has merit but does not fully meet PLOS ONE’s publication criteria as it currently stands. Therefore, we invite you to submit a revised version of the manuscript that addresses the points raised during the review process.

We look forward to receiving your revised manuscript.

Kind regards,

Asma Haque, Ph. D

Academic Editor

PLOS ONE

Journal Requirements:

3. Please upload a copy of all Figures, to which you refer in your manuscript or PDF file. If the figure is no longer to be included as part of the submission please remove all reference to it within the text.

4. Please upload a copy of Supporting Information Figure/Table/etc. S1 Figure, S2 Figure, S1 Table and S2 Table which you refer to in your text on pages 14 and 15.

Reviewers' comments:

Reviewer's Responses to Questions

**Comments to the Author**

1. Is the manuscript technically sound, and do the data support the conclusions?

Reviewer #1: Yes

Reviewer #2: No

2. Has the statistical analysis been performed appropriately and rigorously? 

Reviewer #1: Yes

Reviewer #2: No

3. Have the authors made all data underlying the findings in their manuscript fully available?

Reviewer #1: Yes

Reviewer #2: Yes

4. Is the manuscript presented in an intelligible fashion and written in standard English?

Reviewer #1: Yes

Reviewer #2: Yes

5. Review Comments to the Author

Reviewer #1: Title

The title provides a clear and concise summary of the topic and scope of the study. It effectively communicates the main focus of the research, which is to investigate the correlation between protein expression and sequence-related features of mRNA and protein in Escherichia coli K-12 MG1655 model. The title also specifies the organism and model used in the study, which adds to its specificity and relevance. Overall, the title is well-written and effectively conveys the main topic and scope of the research.

Material and Method

Workflow

In work flow section "While simultaneously used" should be changed to "while also being used" or "while being used simultaneously."

Expression data

"Taniguchi et al." should be referred to as [11].

Calculating translating rate

It might be helpful to provide additional context or information to fully understand the significance of calculating the initiation rate, elongation rate, and translation rate using the Transim model.

Discussion

Then, our research evaluated the correlation of each of these features with the amount of PA/mRNA and found that the correlation of translation rate from Transim had a lower Spearman coefficient (R=0.4316) coefficient of the elongation rate (R=0.6538)," the word "coefficient" is repeated unnecessarily.

When considering the whole mRNA sequence, SVR gave the best predictive ability with both PA and PA/mRNA outputs, reaching Spearman coefficients of RPA=0.6699 and RPA/mRNA=0.6536, respectively," the term "Spearman coefficients" should be replaced with "Spearman correlation coefficients”.

“the term "New_TR" is undefined and should be explained earlier in the paragraph.

Reviewer #2: The study presented by the authors is interesting, but the manuscript requires major revisions to improve the precision and simplicity of the sentences. The authors should replace non-scientific language such as "popular" and "great" with appropriate scientific terms. Additionally, the authors should be consistent in their use of algorithm or method names.

Introduction:

The introduction should include references to previous studies to provide relevant literature. The authors should also provide more details about the Gilad Shahman and Tamir Tuller model. In the last paragraph of the introduction, the authors should clearly mention the objects and results and explain the novelty of their work.

In the sentence "The expression of recombinant protein from the microbial host strains can be considered a great step forward in microbiology," the word "great" should be replaced with a more appropriate word.

The sentence "However, there still has not been a convenient optimization method or tools that could" should use the plural form "methods" instead of "method." The word "convenient" should be clarified to reflect its meaning.

The sentence "However, the model's translation rate did not have a sufficient Spearman's rank correlation with protein level (reported to be up to 0,36)" should use a decimal point instead of a comma to represent the correlation coefficient, i.e., "0.36."

Material and methods:

The material and methods section requires serious rearrangement and more details about the data and methods. The authors should describe the dataset before the workflow to enable readers to understand the algorithms' application on the data.

The sentence "The CDS sequences of every gene in E. coli were retrieved from Gene_sequence.txt from the Regulon database" should explain why the Regulon database was preferred over Genbank, provide a reference for the database, and indicate the number of sequences retrieved.

The authors should provide a reference for the Ecocyc database and explain why it was chosen.

The authors collected sequence and expression data from two different resources, and they should clarify the experimental conditions under which the gene expression data was collected. The authors should also acknowledge that their conclusion/finding might be limited to the conditions under which the gene expression data was generated.

The authors should justify their decision to use microarray data instead of bulk RNA seq datasets. Additionally, the authors should indicate whether they used any evaluation measures when generating features and the validation measures used for machine learning models.

In the sentence "Then, the capabilities of typical machine learning algorithms, including Linear Regression (LR), Ridge, Lasso, Elastic net, Random Forest (RF), and Support vector regression (SVR)," the authors should use the same abbreviation pattern throughout the manuscript.

Results:

Although the results are reasonably organized and structured, the technical shortcomings in the methodology could largely affect the results.

I hope these suggestion will help to improve your research contribution. Looking forward to your revisions.

6. PLOS authors have the option to publish the peer review history of their article (what does this mean?). If published, this will include your full peer review and any attached files.

Reviewer #1: No

Reviewer #2: No

---

## [Author Response · Author response to Decision Letter 0]

13 Jun 2023

Jun 12 2023 11:59PM

Comments to the Author

1. Is the manuscript technically sound, and do the data support the conclusions?

Reviewer #1: Yes

Reviewer #2: No

2. Has the statistical analysis been performed appropriately and rigorously?

Reviewer #1: Yes

Reviewer #2: No

3. Have the authors made all data underlying the findings in their manuscript fully available?

Reviewer #1: Yes

Reviewer #2: Yes

4. Is the manuscript presented in an intelligible fashion and written in standard English?

Reviewer #1: Yes

Reviewer #2: Yes

5. Review Comments to the Author

Reviewer #1: Title

The title provides a clear and concise summary of the topic and scope of the study. It effectively communicates the focus of the research, which is to investigate the correlation between protein expression and sequence-related features of mRNA and protein in Escherichia coli K-12 MG1655 model. The title also specifies the organism and model used in the study, which adds to its specificity and relevance. Overall, the title is well-written and effectively conveys the main topic and scope of the research.

Material and Method

Workflow

In work flow section "While simultaneously used" should be changed to "while also being used" or "while being used simultaneously."

Response: Thank you for your correction, we have fixed the issue.

Expression data

"Taniguchi et al." should be referred to as [11].

Response: Thank you for your correction, we have fixed the issue.

Calculating translating rate

It might be helpful to provide additional context or information to fully understand the significance of calculating the initiation rate, elongation rate, and translation rate using the Transim model.

Response: Thank you for your constructive and helpful suggestion. To elaborate the context of calculating translating-reated rates, we have include an explanation: “First, initiation rate, elongation rate, and translation rate were calculated by the Transim model, with a text file consisting of names, mRNA sequences (UTR+CDS), and start site positions, using the recommended format of Transim. Sequences with the RBS too short for the model to calculate the initiation rate were discarded in the final file. Initiation rate and elongation rate data will then be used as the basis input of machine learning models for calculating a new translation rate with higher correlation to the expression level compared to the original Transim’s translation rate.”

Discussion

Then, our research evaluated the correlation of each of these features with the amount of PA/mRNA and found that the correlation of translation rate from Transim had a lower Spearman coefficient (R=0.4316) coefficient of the elongation rate (R=0.6538), the word "coefficient" is repeated unnecessarily.

Response: Thank you, we have corrected the issue.

When considering the whole mRNA sequence, SVR gave the best predictive ability with both PA and PA/mRNA outputs, reaching Spearman coefficients of RPA=0.6699 and RPA/mRNA=0.6536, respectively," the term "Spearman coefficients" should be replaced with "Spearman correlation coefficients”.

Response: Thank you, we have corrected the issue.

“the term "New_TR" is undefined and should be explained earlier in the paragraph.

Response: Thank you for your constructive and helpful suggestion. To clarify the term “New_TR”, we added an explanation in the introduction: the first objective is applying machine learning methods to calculate a new translation rate parameter from Transim’s initiation and elongation rate on different E. coli genes (termed “New_TR”), aiming to improve the original “Transim translation rate”.

Reviewer #2: The study presented by the authors is interesting, but the manuscript requires major revisions to improve the precision and simplicity of the sentences. The authors should replace non-scientific language such as "popular" and "great" with appropriate scientific terms. Additionally, the authors should be consistent in their use of algorithm or method names.

Response: Thank you for your suggestion. Based on that, we have changed “popular machine learning models” to “traditional machine learning models”, “popular microbial host” to “extensively used microbial host”, “a great step forward” to “a major step”

Introduction:

The introduction should include references to previous studies to provide relevant literature. The authors should also provide more details about the Gilad Shahman and Tamir Tuller model. In the last paragraph of the introduction, the authors should clearly mention the objects and results and explain the novelty of their work.

Response: Thank you for your constructive comments. In order to state the novelty of this research, we have included the statement: “From our knowledge, there has not been a machine learning-based model integrating both translating rate-related features (based on Transim), codon types, and protein stability-related features.” 

For the clarification of Gilad Shahman and Tamir Tuller model, we have provided the elaboration “Specifically, the model consists of three calculation step: (1) calculate the translation initiation rate: as the rate at which the ribosome approaches the start codon, this initiation rate is calculated based on mRNA’s interaction with ribosomal RNA; (2) Calculation of translation elongation rate: based on the translation speed of each codon from the ribosomal profiling data to estimate the elongation rate on all codons of a gene; (3) Using the TASEP algorithm with high resolution to simulate the translation using the starting and elongation rates calculated in steps (1) and (2). TASEP provides the ability to predict translation rate, ribosome density, amount of translation termination, and the occurrence of ribosomal jamming. However, the Spearman’s coefficient correlation from this model’s output with the “true” protein abundance was only 0.36 [2].”

In the sentence "The expression of recombinant protein from the microbial host strains can be considered a great step forward in microbiology," the word "great" should be replaced with a more appropriate word.

Response: Thank you for the comment. We have changed it from “great” to “major”.

The sentence "However, there still has not been a convenient optimization method or tools that could" should use the plural form "methods" instead of "method." The word "convenient" should be clarified to reflect its meaning.

Response: Thank you for the comment. We have used a clearer word “ease-of-use” to replace “convenient”

The sentence "However, the model's translation rate did not have a sufficient Spearman's rank correlation with protein level (reported to be up to 0,36)" should use a decimal point instead of a comma to represent the correlation coefficient, i.e., "0.36."

Response: Thank you, we have corrected the error.

Material and methods:

The material and methods section requires serious rearrangement and more details about the data and methods. The authors should describe the dataset before the workflow to enable readers to understand the algorithms' application on the data.

Response: To clarify the microarray dataset, we have described the process of collecting said data: “We processed this data by keeping only the expression profiles of wild-type E. coli MG1655(WT) and removed all the rest containing the mutant strain profiles. For each profile (equivalent to one culture condition), we calculated the mean value of all the replicates. The mean values of all conditions will then be normalized by the quantile normalization method using preprocessCore library in R programming language. Finally, the topic will obtain the mean value (mean) of each gene calculated from all microarray experiments, called mRNA expression data from microarray experiment (referred to as mRNAmicroarray).”

The sentence "The CDS sequences of every gene in E. coli were retrieved from Gene_sequence.txt from the Regulon database" should explain why the Regulon database was preferred over Genbank, provide a reference for the database, and indicate the number of sequences retrieved. 

Response: Thanks for the kind and specific response. We have included the explanation in the Material and methods session: “This is a primary database focusing on E. coli transcriptional regulation, which would ensure the validity and the specificity of the data, compared to general sequence databases such as Genbank [16]”

The authors should provide a reference for the Ecocyc database and explain why it was chosen.

Response: Thanks for the review, we have provided the explanation “Ecocyc database was chosen as it is a manually curated biological database dedicated to E. coli K-12 that has its data synchronized to RegulonDB. Therefore, it is feasible to combine sequential information from the two databases [15].”

The authors collected sequence and expression data from two different resources, and they should clarify the experimental conditions under which the gene expression data was collected. The authors should also acknowledge that their conclusion/finding might be limited to the conditions under which the gene expression data was generated.

Response: Thank you for the suggestion, we have provided an acknowledgement on the shortcoming of our research data. About the experimental condition, we have added the information accordingly “Wild-type E. coli was grown on M9 glucose medium (2 g1-1) under aerobic and anaerobic conditions for gene profiling experiments and the measurements were done in triplicate. We processed this data by keeping only the expression profiles of wild-type E. coli MG1655 (WT) and removed all the rest containing the mutant strain profiles. For each profile (equivalent to one culture condition), we calculated the mean value of all the replicates [23].”

The authors should justify their decision to use microarray data instead of bulk RNA seq datasets. Additionally, the authors should indicate whether they used any evaluation measures when generating features and the validation measures used for machine learning models. 

Response: Thank you for your comment. To clarify the reason behind using microarray data, we have added the explanation: “Microarray has been used as a method to measure mRNA abundance on a genome scale, and thus could be used as an alternative for RNAseq, especially since only protein transcripts were examined in this research.”

For translating-related features, we calculated using the Transim model, and the results were checked for their correlation with protein abundance and protein abundance per mRNA. There were sequences discarded due to not having RBS motifs for calculation of initiation rates.

For measures to evaluate features, we have provided the context of choosing uAUG based on the position of the ORF: “in-frame (corresponding to the ORF), out of frame, and no uAUG found in the 10-base upstream of the start codon in each mRNA sequence”.

In the sentence "Then, the capabilities of typical machine learning algorithms, including Linear Regression (LR), Ridge, Lasso, Elastic net, Random Forest (RF), and Support vector regression (SVR)," the authors should use the same abbreviation pattern throughout the manuscript.

Response: We have fixed the inconsistency in using abbreviation.

Results:

Although the results are reasonably organized and structured, the technical shortcomings in the methodology could largely affect the results.

I hope these suggestion will help to improve your research contribution. Looking forward to your revisions.

6. PLOS authors have the option to publish the peer review history of their article (what does this mean?). If published, this will include your full peer review and any attached files.

Do you want your identity to be public for this peer review? For information about this choice, including consent withdrawal, please see our Privacy Policy.

Reviewer #1: No

Reviewer #2: No

---

## [Editor Report · Decision Letter 1]

28 Jun 2023

Analyzing the correlation between protein expression and sequence-related features of mRNA and protein in Escherichia coli K-12 MG1655 model

Correlation between protein expression and sequential features

PONE-D-23-08391R1

Dear Dr. Nguyen ,

We’re pleased to inform you that your manuscript has been judged scientifically suitable for publication and will be formally accepted for publication once it meets all outstanding technical requirements.

Kind regards,

Asma Haque, Ph. D

Academic Editor

PLOS ONE
---

## [Editor Report · Acceptance letter]

4 Jul 2023

PONE-D-23-08391R1 

Analyzing the correlation between protein expression and sequence-related features of mRNA and protein in *Escherichia coli* K-12 MG1655 model
Correlation between protein expression and sequential features 

Dear Dr. Nguyen:

I'm pleased to inform you that your manuscript has been deemed suitable for publication in PLOS ONE. Congratulations! Your manuscript is now with our production department. 

Kind regards, 

on behalf of

Dr. Asma Haque 

Academic Editor

PLOS ONE